# Chromosome Imbalances in Neuroblastoma—Recent Molecular Insight into Chromosome 1p-deletion, 2p-gain, and 11q-deletion Identifies New Friends and Foes for the Future

**DOI:** 10.3390/cancers13235897

**Published:** 2021-11-24

**Authors:** Jikui Guan, Bengt Hallberg, Ruth H. Palmer

**Affiliations:** 1Department of Pediatric Oncology Surgery, Zhengzhou Key Laboratory of Precise Diagnosis and Treatment of Children’s Malignant Tumors, Children’s Hospital Affiliated to Zhengzhou University, Zhengzhou 450018, China; jikui.guan@gu.se; 2Department of Medical Biochemistry and Cell Biology, Institute of Biomedicine, Sahlgrenska Academy, University of Gothenburg, 40530 Gothenburg, Sweden

**Keywords:** MYCN, ALK, ALKAL2, PHOX2B, DLG2, haploinsufficiency

## Abstract

**Simple Summary:**

Neuroblastoma is a pediatric cancer that arises in the sympathetic nervous system. High-risk neuroblastoma is clinically challenging and identification of novel therapies, particularly those that offer a reduction in morbidity for these patients, is a high priority. Combining genetic analyses with investigation of molecular mechanisms, while considering recent advances in our understanding of key developmental events, provides avenues for future treatment. Here we review and highlight several recently published articles that address novel molecular mechanisms arising from chromosome 1p, 2p, and 11q aberrations, which likely contribute to high-risk neuroblastoma, and discusses their potential impact on treatment options.

**Abstract:**

Neuroblastoma is the most common extracranial solid pediatric tumor, with around 15% childhood cancer-related mortality. High-risk neuroblastomas exhibit a range of genetic, morphological, and clinical heterogeneities, which add complexity to diagnosis and treatment with existing modalities. Identification of novel therapies is a high priority in high-risk neuroblastoma, and the combination of genetic analysis with increased mechanistic understanding—including identification of key signaling and developmental events—provides optimism for the future. This focused review highlights several recent findings concerning chromosomes 1p, 2p, and 11q, which link genetic aberrations with aberrant molecular signaling output. These novel molecular insights contribute important knowledge towards more effective treatment strategies for neuroblastoma.

## 1. Introduction

Neuroblastoma is the most common extracranial solid tumor in children, accounting for approximately 8% of all malignancies in children and 15% of all cancer-related deaths in this population [1]. The median age at diagnosis is around 17 months, with 90% of patients younger than 5 years and diagnosis rare after 10 years of age [2]. Neuroblastoma is a heterogeneous cancer with different prognosis depending on criteria such as age at presentation, spread of disease, and biological and genetic characteristics at diagnosis [3,4,5]. 

Neuroblastoma is considered to originate from undifferentiated neural crest cells and often presents as tumors located in or in close proximity to the adrenal glands or sympathetic ganglia [6,7]. Developmentally, the chromaffin cells of the adrenal medulla arise from migratory neural crest cell derived sympathoadrenal precursor cells (Figure 1). These cells originate from the dorsal neuroepithelium and migrate to the vicinity of the dorsal aorta. Here, neural crest cells differentiate to give rise to sympathetic neurons and chromaffin cells of the adrenal medulla. Lineage-tracing experiments in mice have revealed that chromaffin cells in the adrenal gland arise mostly from embryonic nerve-associated cells of neural crest origin named Schwann cell precursors [8]. Single-cell RNA sequencing of the transcriptome from developing human and mouse adrenal glands, as well as from low versus high risk neuroblastoma samples, has further increased our knowledge of developmental trajectories and cellular states in neuroblastoma [8,9,10,11]. During the developmental transition from Schwann cells, cells pass through a transient cellular state where they exhibit a transcriptional identity known as the ‘bridge cell signature’. Many of the genes found in this ‘bridge cell signature’ are known to be important for differentiation, such as ASCL1, which is a regulator of neuronal differentiation. Recently it was reported that the non-phosphorylated form of ASCL1 mediates mitotic exit and neuronal differentiation [12]. Another ‘bridge cell’ gene is PHOX2B, which is a transcription factor involved in the development of noradrenergic neuron populations, and a known predisposition gene for neuroblastoma. One way in which PHOX2B might contribute to pathogenesis includes via regulation of ALK, as it has been shown to increase ALK expression *in vitro*. PHOX2B is currently employed as a marker for peripheral neuroblastic tumors ([13,14] and refs therein). 

Additionally, other important genetic markers for neuroblastoma are mutations in *ALK*, *PTPN11, TERT*, *ATRX*, and amplification of *MYCN*. In relapsed neuroblastoma cases, mutations are more frequently observed in *ALK*, as well as other well-known oncogenes and tumor suppressor genes, such as *NF1*, *RAS*, and *RAF*, and also *PTPN11* and *FGFR* [13,16]. Furthermore, telomere maintenance mechanisms are adopted by neuroblastoma cells, favoring unlimited proliferation in the presence of genome instability and the chromosomal end replication crises. Telomerase activation is commonly observed in high-risk neuroblastoma, such as *MYCN*-amplified tumors and *TERT*-rearranged tumors. Furthermore, somatic mutations in *ATRX* have recently been reported as being connected with alternative lengthening of telomeres, which has recently been summarized [17]. While genetic mutations are rare in neuroblastoma, tumors often harbor many chromosomal aberrations that add to the complexity of the genetic landscape. Neuroblastoma was first suggested as a gene-dosage disorder in the 90s [18,19], and today the International Neuroblastoma Risk Group (INRG) uses a classification system based on several key criteria that include age, stage, tumor histology, *MYCN* amplification (MNA), 11q-deletion, and ploidy to define very low-, low-, intermediate- and high-risk groups according to 5-year event-free survival (EFS) [20,21]. Low-risk neuroblastomas are mostly observed in children less than 1.5 years old, and often undergo spontaneous differentiation or regression with little or no intervention. Intermediate neuroblastomas are usually treated with chemotherapy and surgery and have a better prognosis than cases that are classified as high-risk neuroblastoma, which also tend to present after 1.5 years [20]. Treatment of high-risk patients includes immunotherapy, radiotherapy, surgery, high dose chemotherapy, autologous stem cell transplantation, and combination treatments. Despite improved clinical treatments, the long-term survival of children and the 5-year survival with high-risk neuroblastoma is less than 50 percent (Table 1) [2,21,22].

High risk neuroblastoma exhibits genetic features that include deletion of chromosome arm 1p, gain of parts of 17q, aneuploidy and amplification of the proto-oncogene *MYCN*, and deletion of parts of chromosome arm 11q (Figure 2). At this point, two high-risk neuroblastoma groups are commonly considered: the *MYCN* amplified (20–25%) group and the more common 11q-deletion (35–45%) group, which together represent currently therapeutically challenging cases. Generally, 11q-deletion and *MYCN* amplification do not co-exist in the same tumor, although rare cases that harbor both 11q deletion and *MYCN* amplification have been described and constitute a very high-risk group of neuroblastoma patients (Table 1 [21]). 

While *MYCN* amplification is conceptually easier to understand at the molecular level, the mechanisms underlying 11q-deletion in neuroblastoma have been elusive. Genetic analyses have given weight to the hypothesis that the contribution of 11q-deletion to neuroblastoma pathogenesis seems to be a result of imbalanced and aberrant cell signaling, which affects key processes—such as cell cycle, growth, differentiation, survival and likely the DNA damage response (DDR) [23]. Understanding these underlying molecular mechanisms, and ultimately placing them in a developmental context is important to understand how neuroblastoma arises and persists despite current aggressive treatment regimes. Here, we discuss recently published reports regarding molecular mechanisms in the 11q-deletion and *MYCN*-amplified high-risk neuroblastoma groups.

## 2. Novel Mechanisms Underlying the Contribution of 1p36 Deletion to MYCN Neuroblastoma

The MYCN transcription factor is a known oncoprotein, which forms complex interactions with other proteins and regulates many different target gene regulatory elements to activate or repress gene expression [24,25]. MYCN controls the expression of genes that regulate cell proliferation or cell cycle progression, maintenance of pluripotency of cells and is significantly involved in organogenesis during embryonic development [26]. *MYCN* is located on chromosome 2p24.3 and is amplified in 20–25% of neuroblastoma cases [27,28,29,30,31]. *MYCN* amplification drives neuroblastoma cell proliferation and promotes an undifferentiated neuroblastoma phenotype, which strongly correlates with worse prognosis [20,22,30,32,33]. *MYCN*-amplification is genetically defined as anything from 9 copies up to 500 copies, which are typically localized in double minutes or homogenously stained regions [34] High-risk *MYCN*-amplified neuroblastomas commonly exhibit loss of heterozygosity at chromosome 1p as well as gain of 17q, with 70% of *MYCN* amplified neuroblastoma associated with 1p36 deletions [24,35].

Previous genetic studies have identified candidate neuroblastoma tumor suppressor genes, including *KIF1Bb, CHD5, miR-34a, ARID1A*, and *CAMTA1* located at 1p36 [36,37,38]. Convincing data have been presented previously for many of these loci in neuroblastoma and will not be discussed further here [39]. Two independent reports recently reassessed the impact of 1p loss of heterozygosity on the development of MYCN-driven neuroblastoma [40,41]. In one study, García-López *et al.* genetically deleted the 1p36 syntenic region in mouse neural crest cells (NCCs), developing a mouse model that recapitulates high-risk 1p36 loss of heterozygosity/*MYCN* amplified neuroblastoma [40]. This 1p36 syntenic deletion, which included the *Arid1a* and *Chd5* tumor suppressors, resulted in increased anchorage independent growth and defects in differentiation. They went on to examine the effects of 1p36 deletion in an elegant approach that asked which genetic deletion event at 1p36 was selected for when Mycn was overexpressed in NCCs. They observed that while loss of the 1p36 syntenic region did not affect tumor penetrance, it did decrease the time to tumor onset on Mycn overexpression [40]. Interestingly, they noted that 1p36 deletion resulted in a less differentiated tumor when compared with Mycn overexpression alone. Their analysis of different candidate 1p36 tumor suppressor genes, such as *Arid1a, Chd5, Camta1*, and *Kif1b*, identified *Arid1a* as the gene with strongest correlation between expression levels and time to tumor onset. Overall, this work teased apart some of the challenges associated with understanding 1p36 loss of heterozygosity in MYCN-driven neuroblastoma, identifying the SWI/SNF chromatin remodeling complex subunit Arid1a as a MYCN collaborator that accelerates neuroblastoma onset [40] (Figure 2). 

An independent study of the role of ARID1A in zebrafish resulted in similar conclusions [41]. Here, Shi *et al.* focused directly on *ARID1A* due to its high mutation rate across human cancers, including neuroblastoma [42,43]. They showed that deletion of the zebrafish *ARID1A* homologs, *arid1aa* and *arid1ab*, resulted in accelerated onset and increased penetrance of MYCN-driven neuroblastoma in a transgenic zebrafish model [41]. They further interrogated the role of ARID1A in human neuroblastoma cells, showing that genetic removal of *ARID1A* did not affect proliferation, but did promote cell invasion and migration. Transcriptomic analyses further allowed the authors to identify an adrenergic-to-mesenchymal transition, with neuroblastoma cells lacking ARID1A exhibiting an enriched mesenchymal gene signature [41]. 

Taken together, these reports support and suggest a role for the 1p36 locus and *ARID1A*, in neuroblastoma tumor development, where loss of *ARID1A* modulates onset and invasiveness and induces a mesenchymal tumor phenotype that might increase resistance to chemotherapy (Figure 2). Thus, *ARID1A* now joins the list of candidate tumor suppressor genes present at 1p36 that have potential to impact on neuroblastoma initiation and progression. 

## 3. Working Alone or in Cooperation? 2p-gain and the *ALKAL2, MYCN*, and *ALK* Troika

In vertebrates, activation of the Anaplastic Lymphoma Kinase (ALK) Receptor Tyrosine Kinase (RTK) is regulated by the ALKAL ligands, which drive downstream signal transduction that includes activation of PI3K/AKT and RAS/MAPK pathways [44,45,46,47,48]. In neuroblastoma, activating mutations in ALK are found in both familial and sporadic forms, where they are now appreciated to be present in around 10% of primary cases [49,50,51,52,53]. Activating ALK mutations are associated with worse prognosis in the intermediate- and high-risk groups, and are also more commonly observed in relapsed cases [3,16,54,55]. Several groups have shown that ALK gain-of-function mutations are unable to drive neuroblastoma alone when ALK is expressed at endogenous levels [46,56,57,58]. However, one report, in which ALK is driven in the mouse neural crest by either DBHiCre or TH-IRES-Cre, resulted in tumor development, highlighting the need to consider ALK expression levels carefully in a temporal and spatial context in neuroblastoma [59]. While neither mice nor zebrafish models in which Alk has been activated by in locus knock-in events develop neuroblastoma, they do exhibit phenotypes in neural crest derived structures. For example, an enlarged sympathetic ganglia phenotype is observed with a constitutively active Alk in mice [46,57,58], while in zebrafish, activation of the ALK family RTK Ltk (*Ltk^moonstone^*), results in production of ectopic neural crest derived iridiphore pigment cells [45,56]. 

It is now well accepted that ALK and MYCN cooperate to drive neuroblastoma. Mechanistically, we know that *MYCN* transcription is regulated by ALK signaling activity, and conversely, *MYCN* has been shown to drive *ALK* expression [60,61]. Several groups have reported that MYCN and ALK gain-of-function mutations, either in locus or overexpressed, cooperate to drive transformation, and result in increased neuroblastoma penetrance in both cell line—mouse and zebrafish—models [46,57,58,59,60,62,63]. This is in keeping with previous analyses of neuroblastoma cases, which identified a correlation between coexistence of *ALK* activating mutations and *MYCN* amplification in high-risk neuroblastoma with poor prognosis [3]. 

One outstanding question is the potential for misregulation of the ALKAL ligands in the development of neuroblastoma, since ligands for several of the RTKs are known to impact on tumorigenesis, a prime example being the PDGF ligands in human cancer [64]. The original reports of ALKAL activation of ALK noted that ALKAL2 was expressed in the adrenal glands and that its expression was observed in neuroblastoma patient samples [44,45,46,48,65]. Two recent studies have further addressed a potential function of ALKAL2 in neuroblastoma [46,66]. Javanmardi *et al.* homed in on the observation that *ALKAL2* is located on chromosome 2p25.3 near the end of the chromosome, in close proximity to *ALK* and *MYCN* on chromosome 2p (Figure 3). Thus, all three loci are in the area of ‘2p-gain’, which genetically has been reported to predict poor EFS in neuroblastoma patients [67]. Moreover, there are several reports of congenital neuroblastoma in patients with germline partial trisomy of chromosome 2p [68,69,70,71,72]. Javanmardi *et al.* investigated 365 neuroblastoma cases, showing that approximately 30% exhibited 2p-gain (arbitrarily set as more than 4× the normal copy number) or amplification (more than eight copies) in this region [66] and references therein. Furthermore, they showed that neuroblastoma cell lines express ALKAL2 protein, identifying a potential autocrine/paracrine activation of ALK in a tumor setting.

What happens in neuroblastoma when the troika of ALKAL2, ALK, and MYCN are aberrantly expressed, or activated, in either 2p-gain or amplification? Work from Borenäs *et al.* offers some insight into this question [46]. They reported that expression of the ALKAL2 ligand cooperates with MYCN to drive highly penetrant and aggressive neuroblastoma growth in mice, in the absence of ALK mutation (Figure 3). These tumors exhibited a RNA expression signature similar to that of ALK mutant/MYCN driven neuroblastoma. Importantly, they could also show that these tumors were sensitive to ALK TKI treatment [46]. This finding suggests that aberrant regulation of the ALKAL2, ALK, and MYCN troika in neuroblastoma, for example in 2p-gain, may drive ALK signaling activity and therefore could benefit from ALK TKI treatment.

## 4. New Candidate Tumor Suppressor Genes and Molecular Mechanisms Shed Light on 11q Deletion Neuroblastoma

An elegant cell biology experiment performed in the 1990s showed that genetic transfer of a portion of chromosome 11 (from pter to 11q22.2) consistently caused neuroblastoma cells to differentiate and blocked their ability to proliferate [73]. This dramatic finding was the first hint that this chromosomal region harbored genes important for differentiation [73]. Shortly after this, it was reported that neuroblastomas often exhibit deletion of parts of chromosome 11 [74]. Patients with 11q-deletion represent 35–45% of neuroblastoma, presenting at an older age and correlating with a more aggressive disease stage [75,76,77]. It is common that many patients with 11q-deletion (~45% of 11q-deletions) exhibit a loss of chromosome 11 in its entirety [21]. The remainder of patients with 11q-deletion (~55%) have partial loss of 11q genetic material [21], often starting at 11q13–14 and continuing out to 11qter, at the end of the chromosome. It is interesting that homozygous 11q loss has never been reported, suggesting that the tumor suppressor effect of 11q-deletion in neuroblastoma is haplosufficient. Haploinsufficiency in humans can result in a wide range of clinical outcomes, one of which is development of cancer which can occur if cell growth control or DNA repair genes are mutated [78]. While understanding haploinsufficiency, as well as potential for epigenetics to affect outcome in this context, is challenging in human cancer, a number of mouse models have reported haploinsufficiency as contributing to tumor development. These include, for example, loss of the Hint1 protein in mammary carcinogenesis, Dmp1 loss in Myc-induced lymphomas and p27Kip1 acceleration of a range of tumors [79,80,81]. In a transgenic MYCN neuroblastoma zebrafish model, tumors develop with increased penetrance when one allele of both *arid1aa* and *arid1ab* are absent, indicating a haploinsufficient effect [41]. Similarly, loss of 11q in human neuroblastoma may represent an additional example of haploinsufficiency responsible for increased tumor development, although the genes responsible have been difficult to pinpoint [23]. 

Over the years, several candidate genes, including *CADM1, H2AFX, ATM, CHK1, MRE11*, and *CCND1* that lie within the chromosome 11q-deletion area, have been suggested as either driver mutations or tumor suppressor genes [23] (Figure 4). Many of these candidates fall into the categories of cellular growth control regulators or DNA repair genes that are known to contribute to tumor development in the context of haploinsufficiency [78]. In some of these cases, reduced cell proliferation, cell cycle arrest and neuronal differentiation in cell line experiments has further supported such roles [82,83,84,85]. However, no single candidate 11q gene has been confirmed to play a key role in initiation and progression of neuroblastoma, despite intensive investigation mapping of this deleted region (11q14–11qter) [23,31,74,75,86,87,88].

Three recent articles shed new light on the role of 11q-deletion in neuroblastoma from slightly different angles [21,89,90]. Two groups examined somatic structural variation using whole genome sequence and single-nucleotide polymorphism analysis of neuroblastoma samples from their own samples or from publicly available databases [89,90], while Siaw *et al.* employed a combination of genetics from patient neuroblastomas and developmental cell biology approaches [21]. The work of Lopez *et al.* analyzed a set of neuroblastoma tumors with 11q deletion that led them to propose that disruption and deregulation of the *SHANK2* gene at chromosome 11q13 promotes an undifferentiated neuroblastoma state (Figure 4). *SHANK2*, which encodes a post-synaptic protein, is disrupted in 14% of 11q-deleted high-risk neuroblastoma. Further, overexpression of SHANK2 resulted in inhibition of neuroblastoma cell growth, as well as increased differentiation upon treatment with retinoic acid. Retinoic acid is a known differentiation factor in neuronal cells that is employed as a maintenance therapy in high-risk neuroblastoma [91,92]. Keane *et al.* employed public microarray data, concluding that *DLG2*, located on chromosome 11q14, correlated positively with better prognosis and went on to show decreased cell proliferation in 11q-deleted cell lines upon *DLG2* overexpression. A third report, by Siaw *et al.*, combined genetic data from 11q-abberant neuroblastoma tumors with computational analysis that correlated genes expressed during neural crest development together with prognosis in neuroblastoma, also identifying *DLG2* as a tumor suppressor gene in neuroblastoma (Figure 4). The approach of Siaw *et al.*, built on the developmental findings of Furlan *et al.* [8], hypothesizing that genes included in the ‘bridge cell signature’ would be important for the differentiation of Schwann cell precursor to adrenal chromaffin cells (Figure 1). Indeed, in the bridge signature, expression of *DLG2*, is upregulated in the final step of the Schwann cell precursor to adrenal chromaffin cell differentiation process [8,21]. Supporting this, Siaw *et al.* were able to show that overexpression of DLG2 in neuroblastoma cell lines reduced their proliferation and led to an increased transcription of adrenal specific genes, promoting differentiation to a more chromaffin-like cell type. In an interesting twist, they could also show that ALK/MAPK signaling suppressed *DLG2* transcription, suggesting that oncogenic ALK/MAPK activity blocks differentiation cues through regulation of key components such as DLG2. This mechanistic explanation of a role for *DLG2* as a tumor suppressor was accompanied by a comprehensive genetic analysis of 120 11q-deleted neuroblastoma cases that was able to identify *DLG2* as the most proximal gene in the shortest region of genetic overlap in patient material, with *DLG2* disrupted in all 11q deleted cases examined [21]. These observations led Siaw *et al.* to conclude that *DLG2* acts as a tumor suppressor gene in 11q-deletion neuroblastoma, playing a critical role in the differentiation of neural crest lineages. Interestingly, both SHANK2 and DLG2 are post-synaptic anchor proteins that are both able to modulate transcriptional outcome in neuroblastoma cells from proliferation cues to differentiation cues. How this occurs is not presently understood, but is certainly worthy of future investigation. Together, these three articles emphasize the importance of understanding differentiation mechanisms in a normal cellular circumstances, allowing a better understanding of the perturbations in developmental events that can arise as a result of 11q-deletion events. Indeed, such events in 11q-deletion can be further impacted on by deregulated oncogenic activities, such as *MYCN* amplification and aberrant ALK/MAPK signaling, resulting in normal cell development veering off-track and promoting evolution of high-risk neuroblastoma (Figure 4).

## 5. Friends and Foes—Implications for the Future

In recent years, a better understanding of developmental processes has solidified our understanding of neuroblastoma as a cancer arising from abnormal development of neural crest derived tissues. Key findings, such as the impact of mesenchymal or adrenergic cellular identity, as well as definition of key developmental events by lineage tracing, have provided important insight [8,93]. *MYCN* amplification remains one of the most important prognostic biomarkers in neuroblastoma. However, there is currently much focus on copy-number-variation, whole or segmental deletions, gains, and amplification of chromosomes in neuroblastoma. Such ‘ploidy’ disturbances, arising from gain or loss of segment or individual chromosomes can lead to aberrant protein levels, resulting in increased or decreased activity output from key signaling pathways, resulting in cellular stress that can promote transformation and tumor development [94,95]. In high-risk neuroblastoma, 2p-gain and 11q-deletion cases present developing tumors with aneuploidy challenges. What implications can we draw from the recent reports discussed here?

The reports from Javanmardi *et al.* and Borenäs *et al.* consider 2p-gain and the ALKAL2 ligand, highlighting the potential importance of addressing the activation status of ALK at the protein level in neuroblastoma [46,66] (Figure 5). Future phosphor/proteomics analyses may increase our ability to identify ALK-driven neuroblastomas that may respond to ALK TKI therapy. Here, evaluation of ALKAL2 levels in neuroblastoma could provide a potential biomarker for ALK signaling activity that is not captured by genetic mutation analysis. Furthermore, regarding mutation of ALK at the genomic level, there is currently general agreement regarding ALK activating mutations, particularly in the case of well-characterized kinase domain mutations. Although rare, there has been little investigation of ALK extracellular mutations or other *ALK* aberrations in patient samples, although several neuroblastoma derived cell lines are known to harbor ALK extracellular domain variants that are activating [96,97,98]. Indeed, many RTKs—such as KIT, RET, and EGFR—are activated by mutation of the extracellular domain as well as the intracellular kinase domain [99]. A recent structural study has revealed how the ALKAL ligands interact with the extracellular domain of the ALK receptor [100], offering potential for therapeutic exploitation. Further mechanistic investigation taking this information into account should extend our understanding of the importance of ALKAL ligands for ALK activity and signaling in a neuroblastoma context. 

Together, the findings of Lopez *et al.*, Siaw *et al.*, and Keane *et al.* provide novel genetic and mechanistic insight into the complexities of 11q-deletion in neuroblastoma [21,89,90] (Figure 5). In addition to considering the loss of DDR factors, which decrease genomic integrity [101], it is clear that tumor suppressors, such as DLG2, are also influenced by oncogenic signaling activity. Hypothetically, any oncogenic activation that results in ERK pathway activity, such as ALK activation, can result in repression of DLG2 levels, tipping the balance towards mesenchymal identity and inhibiting differentiation. If this is the case in a neuroblastoma patient tumor setting, then some 11q deletions could potentially respond to ALK or ERK pathway inhibition, perhaps in combination with retinoic acid to promote differentiation (Figure 5).

Another grey area in our understanding is the interplay between copy number and epigenetic modification and resulting gene expression in neuroblastoma. Epigenetics play a critical role in the regulation of temporal and spatial gene expression, including that of oncogenes and tumor suppressor genes, and how these events integrate into the pathology of neuroblastoma is currently unclear [102]. Genetic aberrations in chromatin-modifying proteins are known to impart changes to the epigenetic landscape, promoting transcriptional programs that may be beneficial for initiation and promotion of neuroblastoma. Some well-known chromatin remodeling proteins include ARID1A/B and ATRX, which is one of the few known mutational targets in neuroblastoma [103]. The work of García-López *et al.* and Shi *et al.* implicate *ARID1A*, at 1p36, in neuroblastoma development, although the implications for the overall epigenetic landscape are unclear [40,41] (Figure 5). At this point, we have no epigenetic biomarker that can be employed in neuroblastoma diagnosis and prognosis. However, the work of Shi *et al.* suggests that high-risk neuroblastoma with deletion of 1p36 could exhibit a mesenchymal gene signature that may be exploited to identify increased resistance to chemotherapy [41]. If so, current clinical genetics analyses could be complemented by such a gene signature analysis to aid in future clinical treatment decisions. 

## 6. Conclusions

Taken together, these recent findings shed light on the complexity of genetic imbalance that is found in neuroblastoma. How to best interpret the various chromosomal aberrations, and indeed the different combinations of them, particularly in high-risk neuroblastoma cases, is a key challenge for the field in coming years. However, increased mechanistic understanding, including identification of improper signaling and developmental events, in combination with advanced techniques and novel therapeutic options provides optimism for the future. 

## Figures and Tables

**Figure 1 cancers-13-05897-f001:**
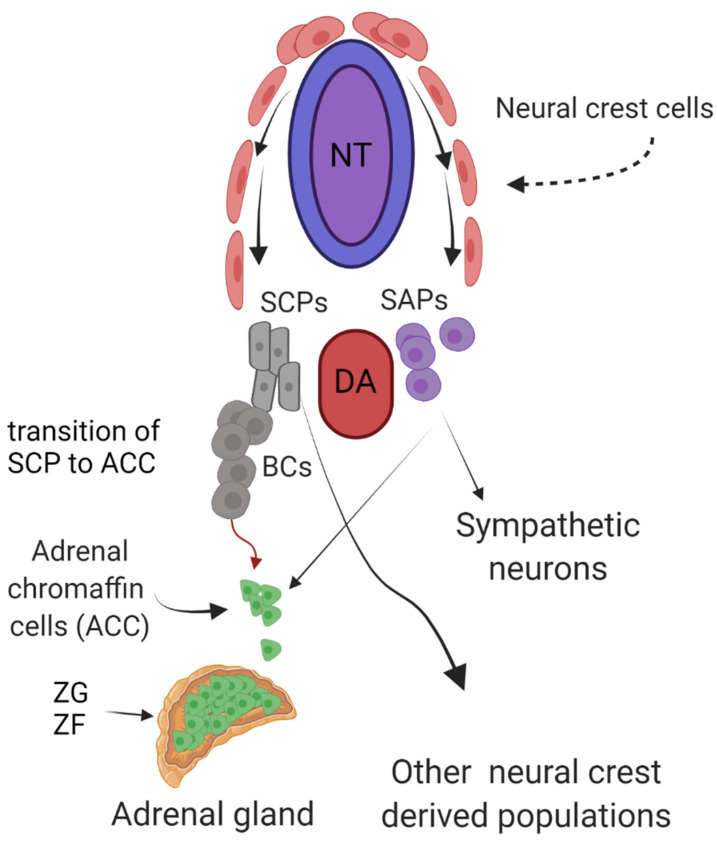
Development of adrenal chromaffin cells (ACCs) from sympathoadrenal precursors (SAPs) and Schwann cell precursors (SCPs). Neural crest cells migrate from the neural tube (NT) to the dorsal aorta (DA). At the DA, SAPs differentiate into sympathetic neurons and ACCs. SCPs migrate on axons to preganglionic neurons that innervate the adrenal gland or differentiate to other neural crest derived populations, such as endoneurial fibroblast, melanocytes, parasympathetic neurons, tooth pulp cells, odontoblasts, chondrocytes, osteoblast, and enteric neurons [15]. ZG, zona glomerulosa; ZF, zona fasciculata; BCs, bridge cells.

**Figure 2 cancers-13-05897-f002:**
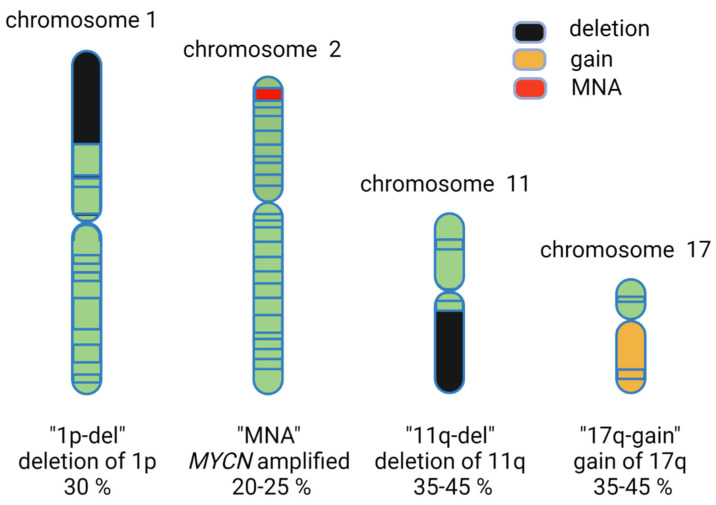
Common known genetic features of neuroblastoma including deletion of chromosome arm 1p (‘1p-del’), gain of parts of 17q (‘17q-gain’), amplification of MYCN (‘MNA’), and deletion of 11q (‘11q-del’).

**Figure 3 cancers-13-05897-f003:**
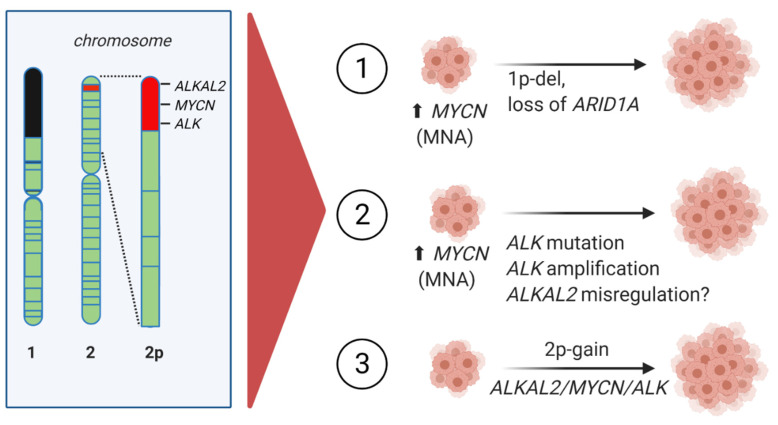
Chromosome 2 chromosomal aberrations involving *MYCN* in high-risk NB. *ALKAL2*, *MYCN*, and *ALK* are localized at chromosome 2p25.3, 2p24.3, and 2p23.2, respectively. (1) *MYCN* amplification together with loss of 1p36, including *ARID1A*. (2) *MYCN* amplification (MNA) with *ALK* mutation, *ALK* amplification or *ALKAL2* misregulation. (3) Copy number gain of chromosome 2p—2p-gain.

**Figure 4 cancers-13-05897-f004:**
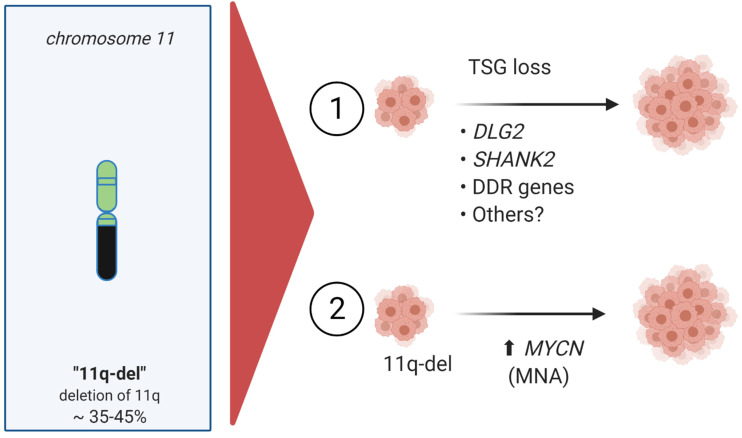
High-risk neuroblastoma with 11q-deletion/non-*MYCN*-amplification (MNA). (1) 11q-deletion tumor suppressor gene (TSG) loss, including genes such as *DLG2*, Discs Large 2; *SHANK2*, SH3 and Multiple Ankyrin Repeat Domain 2; *DDR*, DNA Damage Response genes; Others, including *ATM, CADM1, CHK1, MRE11*, and *CCND1*. (2) 11q-deletion tumor suppressor gene loss combined with *MYCN*-amplification (MNA) comprises are rare, but very high-risk neuroblastoma group.

**Figure 5 cancers-13-05897-f005:**
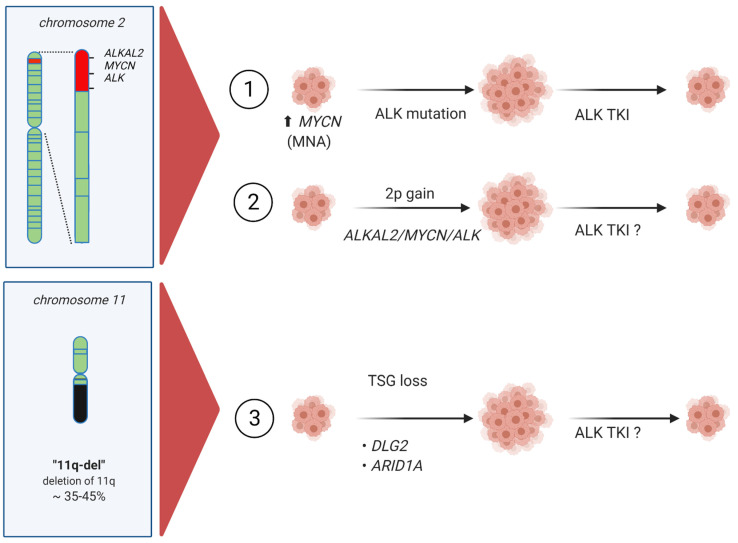
New scenarios for consideration in high-risk neuroblastoma and their treatment. (1) Well established *MYCN* amplification with and without ALK gain-of-function mutations. (2) 2p-gain including *ALKAL2, ALK* and *MYCN*. (3) Tumor suppressor gene (TSG) loss potentially resulting in increased ALK, other RTK or MAPK signaling activity.

**Table 1 cancers-13-05897-t001:** Five years survival after diagnosis based on genomic profile. Overview of commonly employed genetic aberrations in neuroblastoma. Data is accumulated from >400 Swedish neuroblastoma patients [21]. MNA, *MYCN*-amplified; 11q-, 11q deletion; 17q+, 17q gain.

Genomic Profile	Survival 5 Years after Diagnosis
MNA	46%
11q-	48%
MNA and 11q-	0%
17q+ (without MNA or 11q-)	66%
Other segmental	92%
Numerical only	95%

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
