# Peer review of "Chromosome Imbalances in Neuroblastoma—Recent Molecular Insight into Chromosome 1p-deletion, 2p-gain, and 11q-deletion Identifies New Friends and Foes for the Future"

_cancers, 2021, doi:10.3390/cancers13235897_

Round 1

Reviewer 1 Report

Guan et al. summarize and discuss the last findings regarding chromosome imbalances and rearrangements in neuroblastoma, with a special focus on 2p and 11q regions. The authors have qualified expertise in the field, and present a balanced and well-written review. Figures are informative and of good quality, providing a good vision of the author’s ideas. In summary, the review is timely and presents interesting points of view on potential new neuroblastoma combination therapies. Below are some comments mainly intended to improve the organization of the content of the review.

Although the review is mainly focused on 2p and 11q chromosome regions, the authors also include important information on 1p chromosome region that, in opinion of this reviewer, is not properly distinguished in the current manuscript. Section 2 is called “High-risk MYCN neuroblastoma” but instead it is centred on description of anomalies on 1p. For clarity, it is suggested to change the name of section 2 and include a title for this section alluding to the specific content on it (1p and, in a lesser extent, 17q).

Section 3 is focused on 2p alterations, which mainly target MYCN, ALK, and ALKAL2. In line with the previous comment, it is suggested to allude to “2p” in the title of this section.

Figure 3 is focused on chromosome 2, but the part 1 (top schematic numbered 1) includes 1p deletions. To visually clarify this figure, it is suggested to indicate in the part 1 of it the occurring of 1p deletion.

In line 112, the term “1000s” is too colloquial in opinion of this reviewer. Maybe change by “many different” or a similar expression?

Author Response

We thank the reviewer for the valuable comments. Please see the attachment for our response.

Reviewer 2 Report

In this manuscript, Guan et al. review implications of 2p and 11q chromosomal structural variations on neuroblastoma pathogenesis. The review is well structured, the stye fluent and the text pleasant to read. The title of the review fits well to the content of the main text. However, the “simple summary” as well as the abstract is somehow misleading, as these both postulate a review on high-risk neuroblastoma in a more general way. The main text clearly focuses on chromosome 2p and 11q, as predicts the title.

Could the authors please adopt the two paragraphs (simple summary and abstract) that these fit more properly to what is indeed reviewed in the article.

The introduction is more general on neuroblastoma pathogenesis and on what drives high-risk neuroblastoma, which is appropriate in this context. Here the authors also focus on chromosomal structural aberrations, but also describe different somatic mutations like PTPN11, EGFR, NF1 etc. . Telomere maintenance is pivotal for high-risk neuroblastoma and is the strongest molecular marker of the high-risk cases. This aspect must not be missing in such a general introduction on high-risk neuroblastoma. Although TERT and ATRX mutations are mentioned, the role of telomere maintenance in neuroblastoma is missing completely.

Similarly, in the paragraph on MYCN, the consequence of MYCN amplification, which is upregulation of TERT and activation of telomerase should be mentioned and the respective literature cited properly.

The authors report on chromosome 2p gain and on MYCN amplification. These are two different structural aberrations, however, this difference does not become clear enough in the current manuscript. Could the authors please specify what means MYCN amplification in comparison to 2p gain, with regard to the genomic profile, e.g. copy numbers, but also lengths of genomic regions affected etc. How is the effect of amplification versus gain on the expression of affected genes and which impact on patients outcome do both alterations have?

Beside the aspects mentioned above that should be reviewed more in detail, I think this is a valuable review that is worth publication and will be of interest for the readership of Cancers.

Author Response

(The authors gave the same response as above.)
